# PI 3-Kinase and the Histone Methyl-Transferase KMT2D Collaborate to Induce Arp2/3-Dependent Migration of Mammary Epithelial Cells

**DOI:** 10.3390/cells13100876

**Published:** 2024-05-19

**Authors:** Karina D. Rysenkova, Julia Gaboriaud, Artem I. Fokin, Raphaëlle Toubiana, Alexandre Bense, Camil Mirdass, Mélissa Jin, Minh Chau N. Ho, Elizabeth Glading, Sophie Vacher, Laura Courtois, Ivan Bièche, Alexis M. Gautreau

**Affiliations:** 1Laboratoire de Biologie Structurale de la Cellule, CNRS UMR7654, Ecole Polytechnique, Institut Polytechnique de Paris, 91128 Palaiseau Cedex, France; karina.rysenkova@polytechnique.edu (K.D.R.); julia.gaboriaud@polytechnique.edu (J.G.); artem.fokin@polytechnique.edu (A.I.F.); raphaelle.toubiana@polytechnique.edu (R.T.); alex.bense@yahoo.fr (A.B.); camil.mirdass@laposte.net (C.M.); melissajin.75@gmail.com (M.J.); chau.ho@polytechnique.edu (M.C.N.H.); elizabeth.glading@polytechnique.edu (E.G.); 2Pharmacogenomics Unit, Department of Genetics, Institut Curie, Paris Descartes University, 75005 Paris, France; sophie.vacher@curie.fr (S.V.); laura.courtois@curie.fr (L.C.); ivan.bieche@curie.fr (I.B.)

**Keywords:** KMT2D, MLL2, MLL4, PIK3CA, Arp2/3

## Abstract

Breast cancer develops upon sequential acquisition of driver mutations in mammary epithelial cells; however, how these mutations collaborate to transform normal cells remains unclear in most cases. We aimed to reconstitute this process in a particular case. To this end, we combined the activated form of the PI 3-kinase harboring the H1047R mutation with the inactivation of the histone lysine methyl-transferase KMT2D in the non-tumorigenic human mammary epithelial cell line MCF10A. We found that PI 3-kinase activation promoted cell-cycle progression, especially when growth signals were limiting, as well as cell migration, both in a collective monolayer and as single cells. Furthermore, we showed that KMT2D inactivation had relatively little influence on these processes, except for single-cell migration, which KMT2D inactivation promoted in synergy with PI 3-kinase activation. The combination of these two genetic alterations induced expression of the *ARPC5L* gene that encodes a subunit of the Arp2/3 complex. ARPC5L depletion fully abolished the enhanced migration persistence exhibited by double-mutant cells. Our reconstitution approach in MCF10A has thus revealed both the cell function and the single-cell migration, and the underlying Arp2/3-dependent mechanism, which are synergistically regulated when KMT2D inactivation is combined with the activation of the PI 3-kinase.

## 1. Introduction

Cancer develops when cells start proliferating independently from signals that normally regulate their growth in the body. A solid tumor, a mass that forms in one of our organs, represents the progeny of a single cancer cell and can ultimately interfere with proper functioning of that organ [1]. Most human cancers are derived from epithelial cells and are referred to as carcinomas, the most frequent of which are mammary carcinomas [2]. Breast cancer is not a fatal disease, as long as the tumor is restricted to this non-essential organ. Mammary carcinomas are typically removed by surgery. However, mammary carcinoma cells usually end up disseminating into the organism in the process of metastasis, and seed secondary tumors in other organs, such as the brain, the liver, lungs and bones, where their surgical removal is much more difficult [3]. Moreover, secondary tumors can be numerous and scattered throughout the body. As a consequence, chemotherapy that delivers drugs throughout the body remains the only option to treat patients with metastases.

Tumor sequencing has been a major breakthrough in our understanding of the origin of cancers [4]. Indeed, it showed that some genes were frequently mutated in tumors, thus suggesting their critical role in cancer progression. One of these such genes, *PIK3CA*, encodes the catalytic subunit of the PI 3-kinase α and is frequently mutated in various carcinomas [5], and especially in breast cancer, where it is mutated in about one third of the cases [6]. The vast majority of *PIK3CA* mutations are substitutions clustered at two well-defined positions. Such a profile of mutations in tumors is characteristic of oncogenes, which are activated by specific mutations. Since the mutated oncogene product is rendered constitutively active by the mutation, the mutated allele acts as a dominant allele. PI 3-kinase α is an atypical kinase that phosphorylates a lipid rather than proteins. Upon PI 3-kinase-mediated phosphorylation, the membrane phospholipid phosphatidylinositol (4,5) bisphosphate (PIP2) is converted into phosphatidylinositol (3,4,5) trisphosphate (PIP3), which activates several downstream pathways, ultimately resulting in enhanced cell proliferation, survival and migration [7,8]. Because of its implication in tumorigenesis, the PI 3-kinase enzyme has been the subject of intense investigation to identify chemical inhibitors. One of these inhibitors, BYL719, also known as Alpelisib, is recommended for the treatment of mammary carcinomas, where PI 3-kinase activity is elevated due to *PIK3CA* mutations [9,10].

The histone methyl transferase encoding gene *KMT2D,* also known as *MLL2* and *MLL4*, is also mutated in various cancers [11,12]. In breast cancer, the frequency of *KMT2D* mutations is about 4 to 5% [13,14]. *KMT2D* encodes a large enzyme composed of 5537 amino acids. *KMT2D* mutations are mostly substitutions scattered throughout the length of this long gene. A total of 10–20% of mutations truncate the protein, removing in particular its C-terminal catalytic domain. This pattern of mutations is indicative of tumor-suppressor genes, where the two alleles must be inactivated for tumors to arise. The tumor suppressor KMT2D, which incorporates into a large multiprotein complex of the COMPASS family, catalyzes the methylation of lysine 4 in histone 3, and thus regulates the expression of genes in an epigenetic manner [11,12].

We previously found *KMT2D* mutations associated with *PIK3CA* H1047R mutation in an invasive mammary carcinoma, where these two genes were the only cancer-driving genes mutated in this particular tumor [15]. Association of driver mutations in these two cancer-associated genes is present in 1.5% of breast cancer patients in public databases [14]. To examine the potential synergy of these two cancer-associated genes in the transformation process, we combined the activated form of PIK3CA harboring the H1047R mutation with a biallelic knock-out of *KMT2D* in the non-tumorigenic mammary epithelial cell line MCF10A [16], and studied phenotypic consequences of the two genetic alterations when introduced in isolation or combined.

## 2. Materials and Methods

### 2.1. Cells and Drugs

MCF10A cells were cultured in DMEM/F12 medium supplemented with 5% horse serum, 20 ng/mL epidermal growth factor, 10 µg/mL insulin, 500 ng/mL hydrocortisone, and 100 ng/mL cholera toxin. *PIK3CA* H1047R/+ MCF10A clone (HD 101-011) were obtained from Horizon Discovery Ltd. (Cambridge, UK). *KMT2D* KO and double-mutant MCF10A clones were generated in the laboratory. Cells were incubated at 37 °C in 5% CO_2_. All clones were routinely tested for mycoplasma and found to be negative. BYL719 was purchased from MedChemExpress (HY-15244), dissolved in DMSO and used at 5 µM, if not otherwise stated. The total amount of DMSO in treated or control cultures never exceeded 0.01%.

### 2.2. Antibodies, Western Blot and Immunofluorescence

Antibodies targeting GAPDH (#2597762) were purchased from Invitrogen, E-cadherin (#MABT26), Laminin-5 (#MAB19562), ARPC1A (HPA004334), ARPC1B (HPA004832), and alpha-tubulin (#T9026-2ML) from Sigma-Aldrich (Burlington, MA, USA), ARPC5L (ab169763) from Abcam (Cambridge, UK), ARPC5 (#305011, clone 323) from Synaptic Systems (Göttingen, Germany), ARPC2 (#07-227-I) from Merck Millipore (Burlington, MA, USA), and phospho-AKT1 Ser 473 (#9271S) from Cell Signaling Technology (Danvers, MA, USA).

For Western blot analysis, MCF10A cells were lysed in RIPA buffer (50 mM HEPES, 150 mM NaCl, 10 mM EDTA, 0.1% SDS, and 1% NP-40, 0.5% DOC, pH 7.4). SDS-PAGE was performed using NuPAGE 4–12% Bis-Tris gels (Life Technologies, Gaithersburg, MD, USA). Nitrocellulose membranes were incubated with HRP-conjugated secondary antibodies and developed with SuperSignal West Femto Maximum sensitivity substrate (Thermo Fisher Scientific, Waltham, MA, USA).

For the immunofluorescence staining of 3D structures, anti-mouse conjugated with Alexa Fluor 647 antibodies and anti-rat conjugated with Alexa Fluor 594 secondary antibodies were from Life Technologies. Actin was labeled using Acti-stain, TM 488 phalloidin (#PHDG1-A, Tebubio, Le Perray-en-Yvelines, France).

### 2.3. Cas9-Mediated Knock-Out and siRNA-Mediated Knock-Down

To generate *KMT2D* knock-out (KO), parental MCF10A or *PIK3CA* H1047R cells were transfected with the gRNA that targets the CAGAGACCTCTCCCACATGT sequence in the second exon of *KMT2D* (Thermo Fischer CRISPR1024276_CR). Cells were transfected with crRNA:tracrRNA duplex and the purified Cas9 protein, using Lipofectamine CRISPRMAX™ Cas9 transfection reagent (all reagents from Thermo Fisher Scientific). Cells were then diluted to 0.8 cells/well into 96-well plates. Genomic DNA of clones was screened by PCR using KMT2Dfor primer TCTCATGCGTCTGGTGAGTG and KMT2Drev primer GTAAGGCCCTCAGGGAAACC. Loss of the PciI restriction site in the PCR amplicon revealed the presence of Cas9-generated mutations. In candidate clones, PCR amplicons were sequenced using the Sanger method. In the case of overlapping peaks indicating different alleles, PCR products were cloned using the Zero Blunt PCR Cloning Kit (Thermo Fisher Scientific) and resulting plasmids were sequenced to identify the two alleles.

To obtain ARPC5L knock-down, cells were transfected with ARPC5L siRNAs (ON-TARGET smart pool, Dharmacon, L-014690-02-0005) or control non-targeting siRNAs (D-001810-10-20), using lipofectamine RNAiMax (Thermo Fisher Scientific). Image acquisition started at 48 h post-transfection and finished at 72 h.

### 2.4. qRT-PCR

Total RNA from MCF10A cells was extracted using the NucleoSpin RNA Plus Mini kit for RNA purification (Macherey-Nagel, Düren, Germany, #740984.250). To generate cDNA, 1 μg of total RNA and Superscript II reverse transcriptase (Life Technologies, Inc., Gaithersburg, MD, USA, #18064-071) were used. qRT-PCR was carried out using Power SYBR Green PCR master mix (Thermo Fisher Scientific, Waltham, MA, USA, #4368708) on the QuantStudio 7 Flex Real-Time PCR System (Thermo Fisher Scientific, Waltham, MA, USA). The thermal cycling program was as follows: 95 °C for 10 min for the initial denaturation, then 95 °C for 15 s and 65 °C for 1 min repeated 50 times for template denaturation and primer annealing and extension, respectively. A relative transcript level was calculated using the 2ΔΔCt method. The TBP gene was used as a reference gene for normalization. Nucleotide sequences of the primers used are listed in Appendix A.

### 2.5. Cell Cycle

EdU assays were performed overall as previously described [17]. Briefly, cells were cultivated in the presence of 5% growth factor (GF) containing medium (supplemented with 5% horse serum and 20 ng/mL of the epidermal growth factor [EGF]) and seeded at a density of 10^4^ cells per well in 24-well plates. After adhesion of the cells to the substratum, cells were starved from EGF and serum in the so-called 0% GF media for 32 h and then stimulated for 16 h with fresh medium containing either 0% or 1% GF. EdU was added 1 h before cell fixation in 4% paraformaldehyde. Cells were permeabilized in 0.5% Triton X-100 and then labelled using the Alexa Fluor 488 Click-iT EdU Imaging kit (Thermo Fisher Scientific #C10337). The percentage of cells in the S-phase was determined as the ratio between EdU-positive nuclei and DAPI-stained nuclei.

### 2.6. Three-Dimensional Multicellular Structures

Three-Dimensional multicellular structures were obtained as we have previously described [15]. Briefly, MCF10A cells were seeded on top of gelified Matrigel (CB-40230C, Corning) on a Millicell EZ SLIDE 8-well glass chamber slide (PEZGS0816, Millipore) in a medium containing 4 ng/mL EGF and 1% serum and 2% Matrigel. The medium was changed every 3 days for 21 days. Structures were then fixed in 2% PFA in PBS permeabilized with 0.5% Triton X-100, rinsed with PBS/glycine (130 mM NaCl, 7 mM Na_2_HPO_4_, 3.5 mM NaH_2_PO_4_, 100 mM glycine), blocked in IF buffer (130 mM NaCl, 7 mM Na_2_HPO_4_, 3.5 mM NaH_2_PO_4_, 0.1% bovine serum albumin, 0.2% Triton X-100, 0.05% Tween-20) containing 10% fetal bovine serum (FBS) first and then with IF buffer supplemented with 10% FBS and 20 µg/mL goat anti-Rabbit Fc fragment (111-005-046, Jackson ImmunoResearch). Structures were incubated with primary antibodies in the latter buffer, washed with IF buffer, and then incubated with secondary antibody in IF buffer +10% FBS. Then, the structures were incubated with DAPI (Thermo Fisher Scientific), rinsed with IF buffer, and mounted with Mount Liquid Antifade (Abberior, Göttingen, Germany) and sealed with nail polish. Images of the structures were obtained on an SP8ST-WS confocal microscope equipped with an HC PL APO 40×/1.10 W CS2 water immersion objective, a white light laser, HyD and PMT detectors. Image analysis was performed using the Fiji software (Fiji/ImageJ 2.14.0/1.54f) [18]. Structures were manually outlined at their middle section; their size and aspect ratio were then analyzed. Overview images of multicellular structures at 11 days were acquired using an Olympus CKX53 microscope, equipped with a DP22 camera (Olympus) and DP2- SAL firmware (Olympus). Sizes of multicellular structures were calculated using the Fiji software after manually contouring structures.

### 2.7. Cell Migration Assays

All live imaging experiments were performed using an Axio Observer microscope (Zeiss, Oberkochen, Germany) equipped with a Plan-Apochromat 10×/0.25 air objective, a Hamatsu camera C10600 OrcaR2 and a Pecon Zeiss incubator XL multi S1 RED LS. 

For collective migration, 8 × 10^4^ cells per insert (#80209 from Ibidi) were seeded on an 8-well µ-Slide (#80826 Ibidi) in 1% GF medium (1% horse serum, 4 ng/mL EGF). Images were acquired every 10 min for 24 h after release from the insert. Edge progression was measured as the distance moved by the leading edge perpendicular to the initial edge, for each pixel along the wound. Analysis of time-lapse images was performed using Particle Image Velocimetry (PIV) analysis with a MATLAB PIV script [19].

For single-cell migration, an Ibidi 8-well µ-Slide was coated with 20 μg/mL bovine fibronectin (Sigma) for 1 h at 37 °C and 1000 cells were seeded per well in complete 5% GF medium. Images were acquired every 10 min for 24 h. Cell trajectories were obtained using MTrackJ plugin in the Fiji software only for cells that were freely migrating for 6 h or more. The migration persistence of random migration was based on the autocorrelation of angles between displacement vectors, and was calculated using the DiPer program [20].

### 2.8. Statistical Analysis

Statistical analysis was conducted using GraphPad Prism software (v7.00). Two-way ANOVA followed by a post hoc Tukey’s test was used when data followed a normal distribution. Otherwise, the non-parametric Kruskal–Wallis followed by a post hoc Dunn’s test was used. For migration persistence, statistics were calculated using the R software (R 4.3.2), where directional autocorrelation over time was fit by an exponential decay with a plateau as follows:$$A=\left(1-A_{min}\right)*e^{-\frac{t}{\tau }}+A_{min}$$
where *A* is the autocorrelation, *t* the time interval, *A_min_
*the plateau and *τ* the time constant of decay [21]. The plateau value *A_min_* is set to zero for cell lines in vitro, since they do not display overall directional movement. The time constants *τ* of exponential fits were then compared using one-way ANOVA on non-linear mixed-effect models for each condition. Four levels of statistical significance were distinguished: * *p* < 0.05, ** *p* < 0.01, *** *p* < 0.001, and **** *p* < 0.0001.

## 3. Results

### 3.1. KMT2D KO Does Not Contribute to PI3-Kinase-Dependent Proliferation

We have created a set of four MCF10A cell lines to reconstitute the association between two genetic alterations found in breast cancer, the activation of PI 3-kinase by the H1047R mutation, and the inactivation of *KMT2D*. We have first obtained the knock-in cell line where the H1047R mutation was introduced into one of the two alleles of the *PIK3CA* gene from a commercial source. We experimentally verified that this cell line indeed corresponds to what it is supposed to be by sequencing the two *PIK3CA* alleles (Appendix A). We also verified that constitutive activation results in constitutively phosphorylated AKT1 (Appendix A). We then designed a gRNA that targets the second exon of *KMT2D* and obtained biallelic knock-outs (KOs) by CRISPR/Cas9 from both parental MCF10A and *PIK3CA* H1047R cell lines (Appendix A). This set of four cell lines with the parental MCF10A was subsequently used to investigate the consequences of each genetic alteration and their combination.

We first evaluated cell cycling in limiting the amount of growth factor (GF), since GF-independent cell cycle progression is a hallmark of cell transformation in cancer. In 0% GF, MCF10A cells stopped proliferating. However, *PIK3CA* H1047R, but not *KMT2D* KO cells, were able to cycle in this condition (Figure 1, Appendix A). The combination of *KMT2D* KO with *PIK3CA* H1047R did not affect the enhanced cycling due to PIK3CA H1047R alone. Interestingly, the cell lines that were cycling in 0% GF had no advantage over controls when 1% GF was present in the medium: double-mutant cells, in particular, were cycling significantly less than parental MCF10A cells in 1% GF. Since in 0% GF PI 3-kinase activity appears to control the cell cycle in lines harboring the *PIK3CA* H1047R mutation, we studied cell cycling in the presence of 1 or 5 μM BYL719, which is the specific PI3-kinase α inhibitor approved for breast cancer treatment under the name of Alpelisib. BYL719 stopped cell cycling in the two cell lines harboring the *PIK3CA* H1047R mutation at 0% GF, and decreased cycling in all cell lines in the presence of 1% GF in a dose-dependent manner. We conclude from this first set of experiments that the ability of cells to cycle, and in particular when they should not, in the absence of GF, is determined by PI 3-kinase activity and that the inactivation of the histone methyl-transferase encoded by *KMT2D* has no effect in the proliferation of 2D cultures of MCF10A cells.

To explore this point in greater depth, we challenged cells by cultivating them in 3D Matrigel for up to 3 weeks (in the presence of 1% GF). In these settings, cells develop as multicellular spherical structures. Spheres had various sizes and morphologies among different cultures. Cell lines harboring *PIK3CA* H1047R either alone or in combination with *KMT2D*KO were growing into structures that were significantly bigger and had less-regular morphology than controls at 11 days (Figure 2A). In addition, we noticed the presence of protrusions invading into Matrigel in these two cell lines, but not with MCF10A or *KMT2D*KO cell lines. After 21 days, however, the effect of *PIK3CA* H1047R on structure size and shape regularity was less pronounced when *KMT2D* was also inactivated (Figure 2B), suggesting that the combination of these two driver mutations in double-mutant cells attenuated cell transformation induced by strong PI3-kinase activity. BYL719 treatment decreased the aberrant growth and shape of the multicellular structures formed by the *PIK3CA* H1047R cell line.

### 3.2. Synergistic Activation of Migration Persistence by PIK3CA H1047R and KMT2D Inactivation

Migratory behavior is another feature of transformed cells in addition to aberrant proliferation and growth. Therefore we examined the collective cell migration of the mammary epithelial cell lines we had generated. To this end, we seeded the four cell lines at confluence in inserts and lifted the insert to study how the different epithelial monolayers migrated into the free space. Displacement vectors were obtained using Particle Image Velocimetry (PIV). The *PIK3CA* H1047R and the double-mutant cell lines showed significantly enhanced velocity of the edge compared with parental MCF10A and *KMT2D* KO alone (Figure 3, Appendix A). The PIV method allows for the examination of how the information about free space sensed by the first row of cells is transmitted backwards to the follower cells in the monolayer. The velocity and the order parameter, which indicates the coordinated directional migration toward the free space, were transmitted further backwards in the two cell lines expressing active PI 3-kinase compared with controls. Basal and enhanced backward transmission of velocity and order parameter were in all cases abolished upon treatment with BYL719. In this collective migration assay, as well as in previous proliferation assays, *KMT2D* KO, either alone or in combination with *PIK3CA* H1047R, exerted little, if any, influence.

We reasoned that the histone methyl-transferase KMT2D should modulate gene expression in MCF10A cells. Indeed, the list of genes deregulated by *KMT2D* mutations was previously established [22]. Similarly, *PIK3CA* H1047R affects gene expression, and similar lists were also established [23,24]. We selected in these lists of genes that were deregulated by either *KMT2D* mutation or *PIK3CA* H1047R mutation the genes that might be responsible for the phenotypes of proliferation and migration/invasion that we had observed in the four cell lines. We examined the expression of 12 selected genes using qRT-PCR in the presence or absence of the PI 3-kinase inhibitor BYL719 (Appendix A). Out of twelve, one gene, *ZEB1*, was not detectable in MCF10A cells. Only two out of the eleven remaining genes showed significant variations in the set of four cell lines (Figure 4). These two genes were *PLAUR*, which encodes the urokinase plasminogen activator receptor (uPAR), a glycolipid-anchored protein of the cell surface involved in migration and invasion, and *ARPC5L*, which encodes one of the two paralogous ARPC5 subunits of the cytosolic Arp2/3 complex that polymerizes branched actin, which is instrumental in cell migration and intracellular trafficking. These two genes displayed the same pattern of regulation. Both showed a significantly higher expression level in the double-mutant than in parental MCF10A cells. These two genes, which were previously reported to be upregulated in cells expressing *PIK3CA* H1047R, were indeed upregulated in the *PIK3CA* H1047R MCF10A cell line, even if the difference did not reach statistical significance, but not in the *KMT2D* KO cell line. Deregulated expression of these two genes was the first evidence of a potential synergy between *PIK3CA* H1047R and *KMT2D* KO in our MCF10A cell system.

We next focused on ARPC5L, because the role of the Arp2/3 complex in cell migration has been well-established in MCF10A cells [17,21,25]. We depleted ARPC5L from the four cell lines using pools of siRNAs and evaluated collective migration in these cells using the previous assay, utilizing PIV. Surprisingly, ARPC5L depletion did not affect collective migration either in terms of speed or the directionality of migration that is transmitted backward from the free edge of the monolayer (Figure 5). However, our group has previously shown that Arp2/3 activity was implicated in the migration persistence of single MCF10A cells [17,21,25]. So we decided to assay the random migration of single cells in our four cell lines. In this assay, we compared the effect of ARPC5L depletion to that of depletion of its paralog ARPC5. Both were efficiently depleted (Appendix A). All cell lines that harbored a cancer-associated genetic alteration either alone or in combination displayed increased speed of cell migration, but this increased speed was not affected by the depletion of ARPC5L or ARPC5 (Figure 6 and Appendix A). In contrast, when the persistence of directional migration was examined, the *PIK3CA* H1047R cell line displayed increased persistence, which was even stronger when the PIK3CA mutation was combined with KMT2D KO (Appendix A). Importantly, the increased migration persistence in the double-mutant cell line was abolished when ARPC5L, but not ARPC5, was depleted (Figure 6D and Appendix A). This important result of enhanced ARPC5L-dependent migration persistence was verified using a second double-mutant clone (Appendix A). Moreover, ARPC5L depletion did not affect the basal persistence of parental MCF10A cells. Mean Square Displacement (MSD), which is the surface area covered by migrating cells over time, depends on both speed and directional persistence and reflects the ability of cells to disseminate. As a result, MSD showed a stronger increase in the double-mutant cell line, which was no longer the case when ARPC5L was depleted (Figure 6). These data thus indicate that the synergistic enhancement of migration persistence by *PIK3CA* H1047R and *KMT2D* KO can be attributed to ARPC5L, whose expression is also enhanced by the two genetic alterations.

## 4. Discussion

*PIK3CA* acts as a major oncogene when activated by the H1047R mutation. We observed that PIK3CA H1047R MCF10A cells were still cycling in the absence of growth factors, as previously reported [7,26]. These cells were also shown to form enlarged multicellular structures in 3D Matrigel, with aberrant shapes and signs of invasion, a well-established hallmark of cell transformation [27]. We have performed detailed analysis of the migratory behavior using two protocols for collective monolayers and single cells. In both cases, PI3-kinase activity dramatically enhanced cell migration. In Matrigel-coated Boyden chamber assays, enhanced migration and invasion of cancer cells transformed with PIK3CA mutations were shown to depend on AKT1-mediated phosphorylation of cortactin [8].

In contrast, *KMT2D* appeared as a weak tumor-suppressor gene. *KMT2D* KO cells did not behave differently from parental MCF10A cells in proliferation and in most of the migration assays. The only migration parameter that was affected by *KMT2D* KO was the speed of single migrating cells, which was enhanced by *KMT2D* KO to the same level as that of *PIK3CA* H1047R. Surprisingly, in double-mutant cells, the speed of single migrating cells was not increased further. The only synergy between *PIK3CA* H1047R and *KMT2D* KO was found on the directional persistence of individually migrating cells. *PIK3CA* H1047R, but not *KMT2D* KO, already enhanced migration persistence. However, in double-mutant cells, persistence was further increased, suggesting that KMT2D controls this parameter even if this was not apparent in the single *KMT2D* KO.

Studies reporting the combined actions of driver mutations are indeed less numerous and relatively recent compared to those reporting the phenotypes induced by single driver mutations. Mutations of *PIK3CA* were shown to cooperate with *KRAS* mutations in MCF10A cells to induce anchorage-independent growth in soft agar [28]. Results of these combinations are sometimes counter-intuitive. For example, the *PIK3CA* H1047R mutation antagonizes lung metastases in a mouse model of ErbB2-induced breast cancer [29]. Toska et al. had previously studied the role of *KMT2D* inactivation in cells transformed by active PI 3-kinase and found that KMT2D was mediating the resistance to BYL719 in these cells in a mechanism that depended on the transcription of the estrogen receptor [30]. These results emphasize the importance of studying the synergy between oncogenes and tumor-suppressor genes, not only to understand cancer progression but also to develop the most appropriate therapy.

The synergy we reported here between *PIK3CA* H1047R and *KMT2D* KO on the persistence of single-cell migration was associated with enhanced expression of *ARPC5L*, a gene encoding an Arp2/3 subunit. The PIP3-dependent protein kinase AKT1 was shown to phosphorylate KMT2D and thereby decrease its ability to enhance gene expression [30]. But this phosphorylation event cannot account for the synergy of KMT2D KO with PIK3CA H1047R, since KMT2D should thereby be inactivated when PIK3CA harbors the H1047R mutation. This association of enhanced migration persistence with enhanced *ARPC5L* expression immediately attracted our attention, since we had previously shown that the Arp2/3 complex specifically controls the migration persistence of single cells [17,25,31]. The Arp2/3 complex is the only machinery that polymerizes branched actin. Polymerization of branched actin networks generates the pushing force driving membrane protrusions at the leading edge of migrating cells [32]. The Arp2/3 complex is composed of seven subunits. ARPC5 and ARPC5L are paralogous proteins, and, as a consequence, ARPC5-containing Arp2/3 complexes co-exist with ARPC5L-containing Arp2/3 complexes [33].

ARPC5L-containing Arp2/3 complexes are more active in polymerizing branched actin networks, and stabilized more by cortactin against coronin1-mediated debranching than those containing ARPC5 [34]. The two ARPC5 paralogs were recently shown to have opposite effects on the lamellipodial width of B16-F1 cells, suggesting isoform-specific functions [35]. In MCF10A cells, we repeatedly found that migration persistence of single cells was the best read-out of Arp2/3 activity [17,21,25]. The enhanced migration persistence of double-mutant cells depended on ARPC5L and not on ARPC5. Surprisingly, collective migration of MCF10A cells did not depend on ARPC5L, even though cells in the first row made lamellipodia and those further in the monolayer probably made cryptic lamellipodia below the cells in front of them [36]. We have recently generated “super-migrator” cells with enhanced Arp2/3 activity at the cortex, and these genetic manipulations similarly increased the migration persistence of single cells, but not the collective migration of MCF10A cells [25].

Our reconstitution approach of generating genetic alterations and combining them is a straightforward approach to studying the potential cooperation of cancer-associated genes, which are altered in the same tumors. Here, we have identified a major parameter that is synergistically potentialized by *PIK3CA* H1047R and *KMT2D* KO, the migration persistence of single cells, and found a gene, possibly the gene, *ARPC5L*, whose expression is upregulated by these two genetic alterations and that appears responsible for the synergistic phenotype.

## Figures and Tables

**Figure 1 cells-13-00876-f001:**
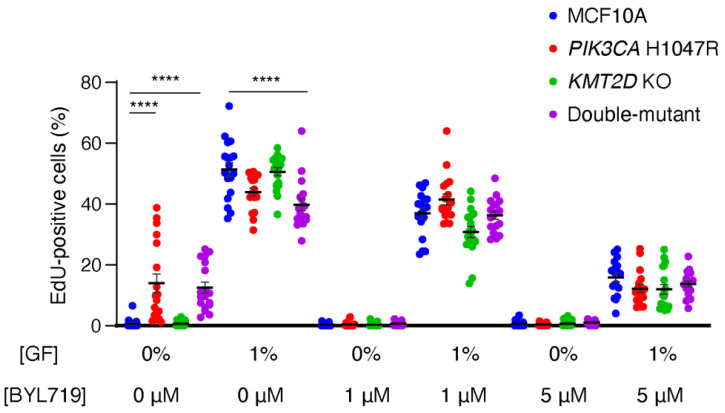
PI 3-kinase activity controls cell cycle progression of MCF10A cells. MCF10A, *PIK3CA H1047R*, *KMT2D KO* and double-mutant cells are cultivated in media containing either 0 or 1% of growth factors (GFs, that is both serum and EGF) and incubated for 1 h in the presence of EdU, which incorporates into DNA during the S-phase. Cells are exposed to 1 µM, 5 µM BYL719 or DMSO as a control vehicle, for 16 h before incubation with EdU. Three independent experiments have been performed, and six fields of view per sample are analyzed (n > 1000 cells). Statistically significant differences with controls are represented by stars: **** *p* < 0.0001 (two-way ANOVA followed by Tukey’s test).

**Figure 2 cells-13-00876-f002:**
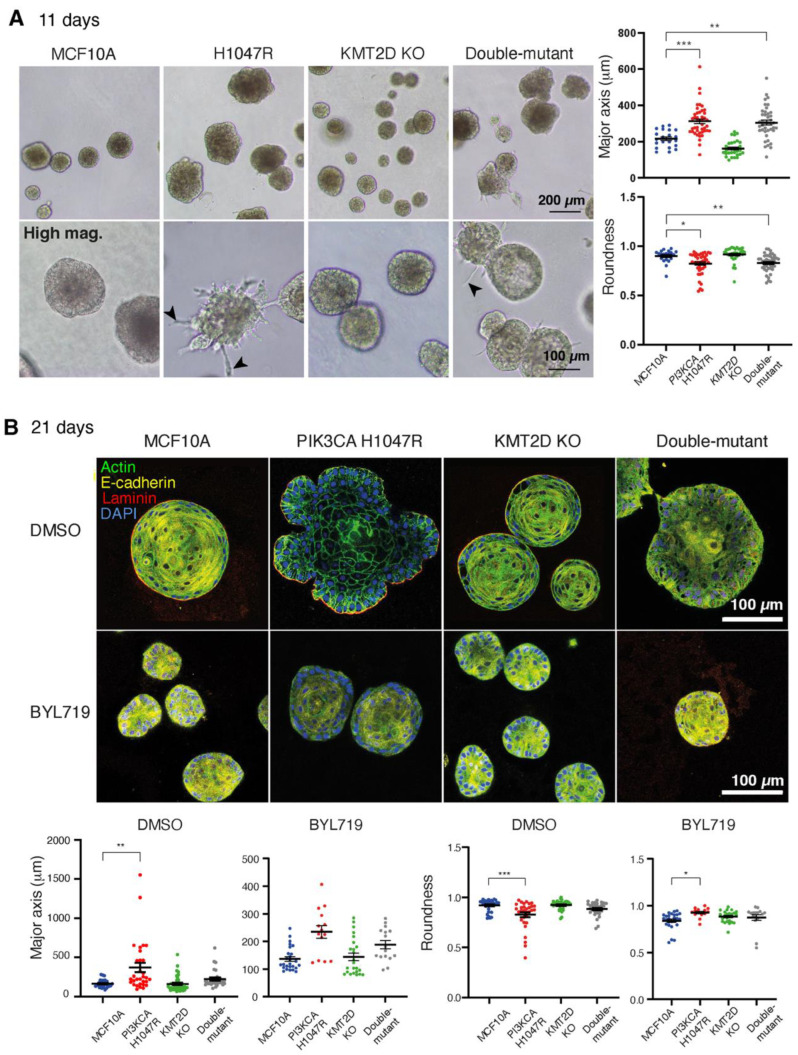
KMT2D inactivation partially corrects the aberrant multicellular structures formed by *PIK3CA* H1047R-expressing cells in Matrigel. (**A**) Morphologies of multicellular structures developed by MCF10A derivatives in Matrigel after 11 days of growth. *PIK3CA* H1047R and double-mutant structures are larger than controls and develop spiky protrusions that invade the Matrigel (arrowheads in high-magnification views). (**B**) Multicellular structures were grown for 21 days in the presence of 1 µM BYL719 or DMSO vehicle. Structures are stained with DAPI (blue), phalloidin (green), E-cadherin (yellow), and Laminin-5 (red) antibodies and their size and shape are quantified using their major axis, which corresponds to the longest diameter of the middle z-section, and their roundness index, which corresponds to the ratio of the minor to the major axis of the same section. Four independent experiments (with DMSO) and two independent experiments (with BYL719) have been performed; more than 20 structures were included for each condition. Data from one representative experiment are displayed. Statistically significant differences with controls are represented by stars: * *p* < 0.05, ** *p* < 0.01, *** *p* < 0.001 (two-way ANOVA followed by Tukey’s test).

**Figure 3 cells-13-00876-f003:**
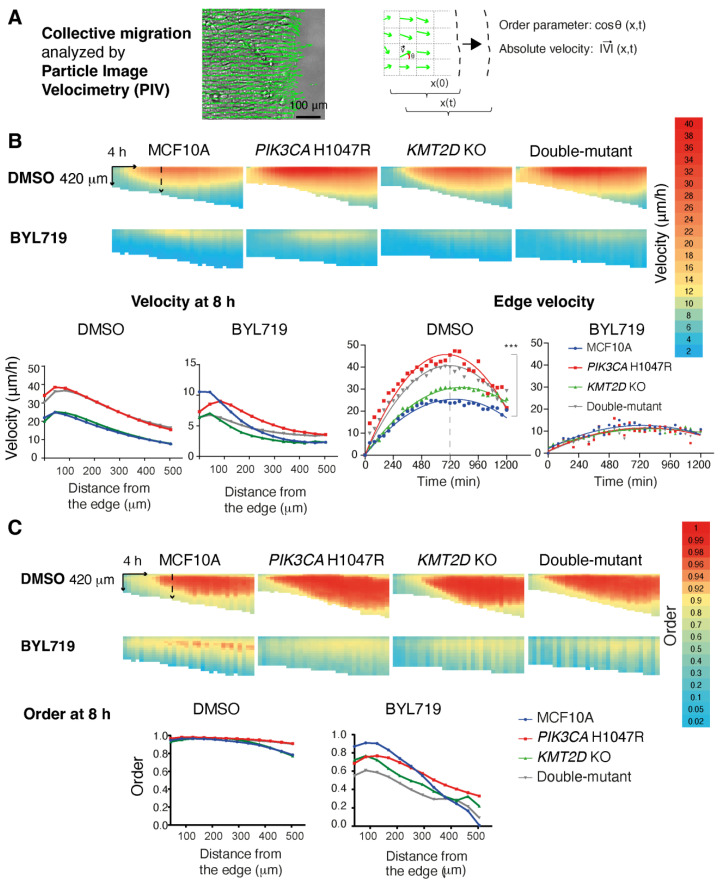
PI 3-kinase activity is critical for collective cell migration. Confluent monolayers of parental, *PIK3CA* H1047R, *KMT2D* KO and double-mutant MCF10A cells are induced to migrate into the free space obtained by lifting a physical boundary. Displacement vectors within the monolayer are obtained using Particle Image Velocimetry ((**A**), PIV). These vectors are analyzed over space and time for their position in terms of their magnitude ((**B**), velocity) and directionality ((**C**), order parameter, i.e., the cosine of the angle made by the displacement vector with respect to the vector oriented toward the free space). These parameters are represented as heat maps. There are three biological repeats with similar results, and four fields of view in each condition. Plots show the average of all measurements. Each pixel of the heatmaps averages all fields of view in all biological replicates over 42 µm and 40 min. These parameters are compared at a single time point or at a single distance from the edge using two-way ANOVA with Tukey’s test.

**Figure 4 cells-13-00876-f004:**
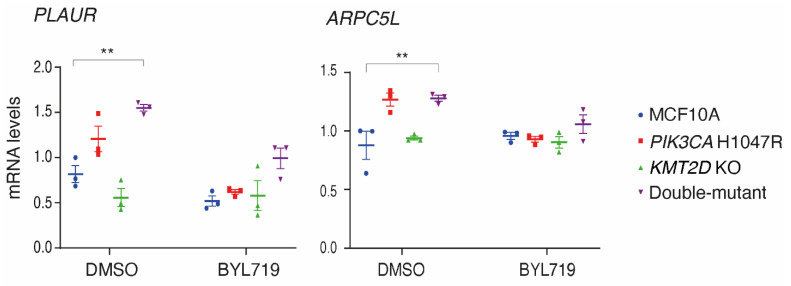
Expression levels of the *PLAUR* gene, encoding the urokinase plasminogen activator receptor (uPAR), and the *ARPC5L* gene, encoding one of the two paralogous ARPC5 subunits of the Arp2/3 complex, are significantly up-regulated in double-mutant cells. qRT-PCR analysis of gene expression is performed on MCF10A, *PIK3CA*H1047R, *KMT2D* KO and double-mutant cells in the presence or absence of 5 µM BYL719. Mean ± SEM, n = 3 biological repeats. Statistically significant differences with controls, using two-way ANOVA followed by Tukey’s test, are represented by stars: ** *p* < 0.01.

**Figure 5 cells-13-00876-f005:**
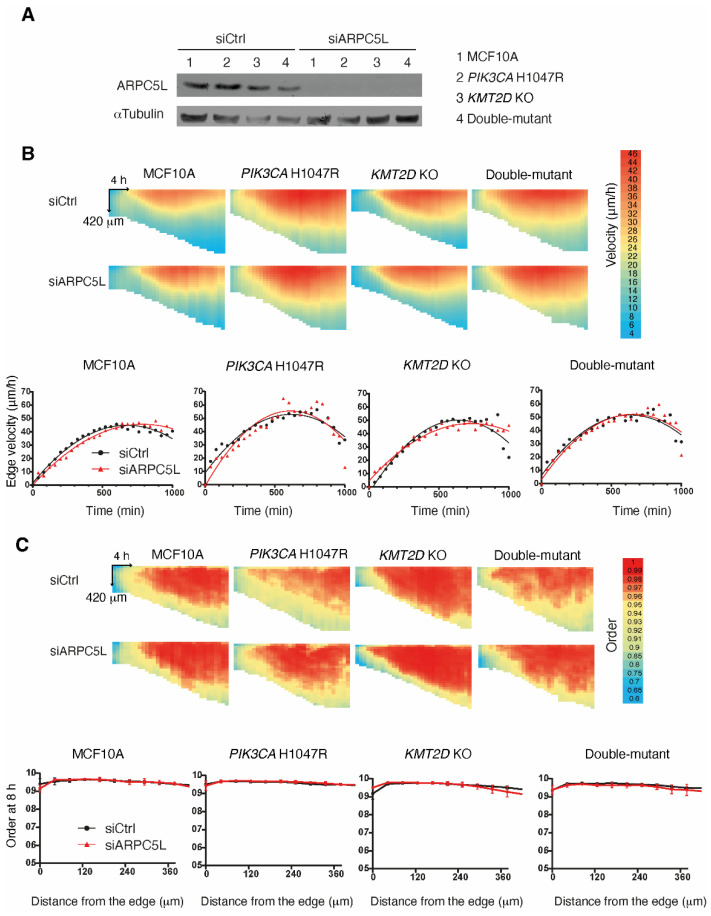
The Arp2/3 subunit ARPC5L does not control collective cell migration. (**A**) ARPC5L is depleted from MCF10A, *PIK3CA* H1047R, *KMT2D* KO and double-mutant cells using siRNA-mediated knock-down. Collective migration is induced by lifting a physical boundary. Displacement vectors within monolayers are obtained using PIV and analyzed as in Figure 3. (**B**) Velocity. (**C**) Order. Two biological repeats with similar results, with four fields of view in each condition. Plots show the average of all measurements. Each pixel of the heatmaps represents an average of all fields of view and all biological replicates over 42 µm and 40 min.

**Figure 6 cells-13-00876-f006:**
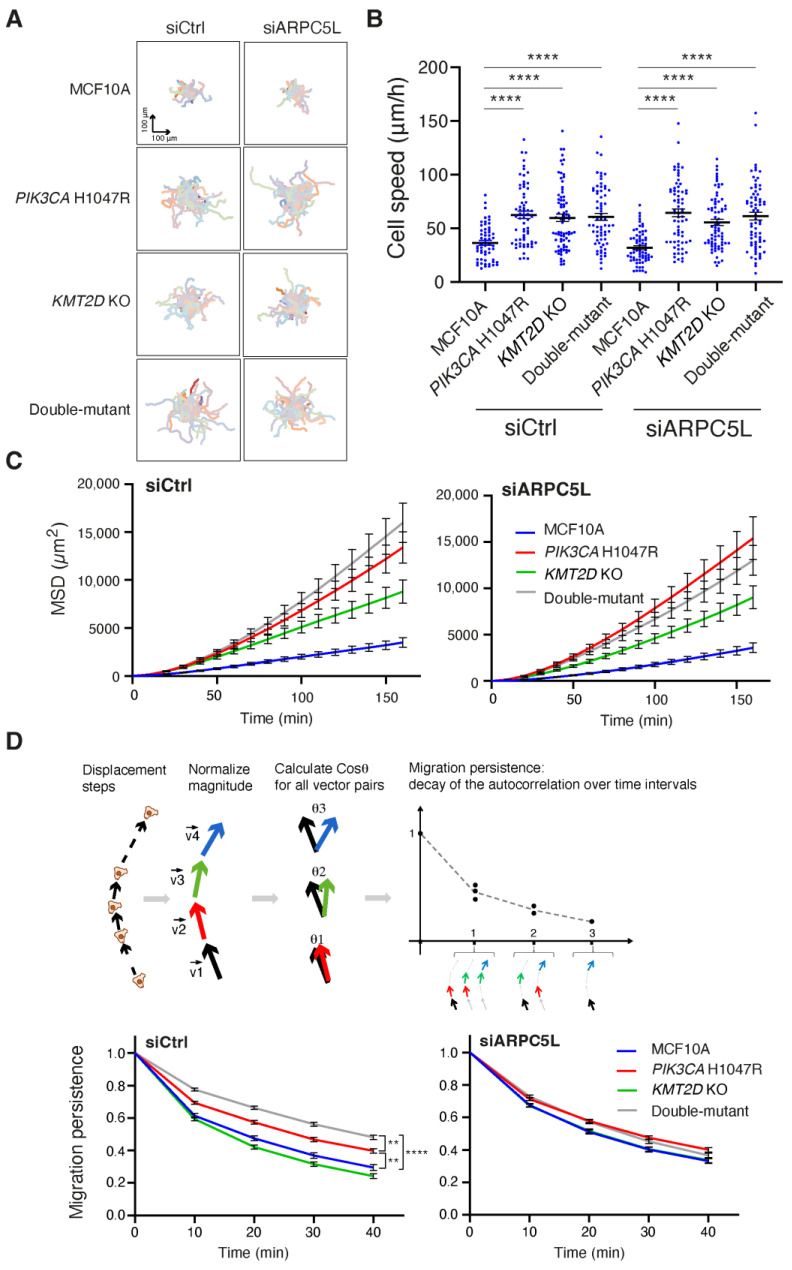
ARPC5L stimulates migration persistence, but not speed, in double-mutant cells. Random migration of single cells is tracked for 6 h. (**A**) Trajectories of 56 MCF10A, *PIK3CA*H1047R, *KMT2D* KO and double-mutant cells, depleted or not depleted of ARPC5L. The origin of each track is registered at the center of the plot. (**B**) Cell speed (mean ± SEM). (**C**) Mean Square Displacement (MSD). (**D**) Migration persistence. Three independent experiments are performed, and the overall number of cells ranges from 56 to 78 for each condition. ** *p* < 0.01, **** *p* < 0.0001 (two-way ANOVA followed by Tukey’s test for speed, 1-way ANOVA on nonlinear mixed-effect models for persistence).

## Data Availability

The MCF10A cell line used in this study was from the collection of breast cell lines organized by Thierry Dubois (Institut Curie, Paris, France).

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
