# Peer review of "PI 3-Kinase and the Histone Methyl-Transferase KMT2D Collaborate to Induce Arp2/3-Dependent Migration of Mammary Epithelial Cells"

_cells, 2024, doi:10.3390/cells13100876_

Round 1

Reviewer 1 Report

Comments and Suggestions for Authors

The authors describe a model system for studying the activity of an activated form of the PI 3-kinase in the presence or absence of histone 17 lysine methyl-transferase KMT2D in a cancer breast cell line. They analyze cell cycle progression and cell migration (collective monolayer and single cells). Because in the double genetically altered cells expression of ARPC5L was increased, they further silenced expression of the encoded protein measuring loss of persistence of directional migration. On the whole the analysis has been done meticulously and the results are convincing. The discussion stays a bit on the beaten path (nothing wrong with that) but I kind of lack ideas on how to proceed further with this elegant model system to get mechanistic insights on cell migration or on whether they expect it to be useful for invasion studies (although MCF10a is non-malignant).

I would have loved to see a comparison of the altered expression of the four cell lines (RnaSeq or proteomics) but this may be a follow-up study on its own and see also also comment 3 below.

Some minor points

1)     Give the composition of the RIPA buffer used or refer to a paper with its composition or indicate the company if it is from a commercial source. Unfortunately, not all lab use exact the same composition of RIPA-buffers.

2)     Correct the tittle on line 146: 2.6 3D instead of 2.6.3  D

3)     In view of the comment above on the incomplete characterization of the expression profile in the four cell lines change in the discussion the sentences on lines 427-430 to something more nuanced.

Here we have identified the a major parameter that is synergistically potentialized by PIK3CA H1047R and KMT2D KO, migration persistence of single cells, and found a gene, possibly the gene,: ARPC5L, whose expression is upregulated by these two genetic alterations and that appears responsible for the synergistic phenotype.

4)     Enlarge panel 6B

5)     Part of suppl. figure S4 is duplicated with main fig 6

Reviewer 2 Report

Comments and Suggestions for Authors

The paper by Rysenkova et al. addresses a relevant question in cancer biology: How do oncogenes and tumor suppressor genes cooperate to produce a cancer phenotype? The overall strategy and experimental work are sound. However, the conclusions are based on an insufficiently established and characterized cellular system.

The genetically modified cells are only characterized by their DNA sequence: Activated PI-3 K should be shown by kinase assays or other methods. KMT2D absence should be shown be a negative Western blot analysis.

Results based on a single cell line created by CRISPR for each construct is not sufficient to deduce the conclusions made.

Several clones for each construct have to be used or further experiments on the cell line should be done confirming that the phenotype is really caused be the oncogene/tumor suppressor gene. In the case of PI-3 kinase the observed phenotype should mimic ectopic expression with activated PI-3 kinase. in case of KMT2D, si experiments could be done giving similar results as the ko experiments.

In conclusion this work can recommended for publication if further experimental work is done.

Author Response

Thanks for your comments. Please find the attached reply.

Reviewer 3 Report

Comments and Suggestions for Authors

1. In the figure 4, mRNA changes in Arpcl5 expression are extremely low. Can the authors explain how they justify rigor? There is no valid to make mechanistic conclusions without further experiments would be need to justify this finding. Consider CHIP.

2. What would be the effect on the metabolic reprogramming in this model? How would that affect your conclusions?

3.Correct for the use of standard gene/protein nomenclature.

Author Response

(The authors gave the same response as above.)

Reviewer 4 Report

Comments and Suggestions for Authors

Karina D. Rysenkova and collaborators describe to cooperation between PIK3CA mutation and KMT2D inactivation for migration of mammary epithelial cells.

PIK3CA gain of function and KMT2D loss induced expression of an Arp2/3 subunit, essential for epithelial cells migration.

My comments are in BOLD.

Figure 1, the ability of cells to cycle, and in particular when they should not, in the absence of GF, is determined by PI 3-kinase. The data is convincing and well performed.

Figure 2, culture in 3D Matrigel for up to 3 weeks (in the presence of 1% GF). H1047R spheres with/out KMT2D KO grown bigger and with irregular morphology at 11 days, the presence of protrusions invading into Matrigel in these two cell lines was apparent, but not with MCF10A or KMT2D KO cell lines.

After 21 days the effect of PIK3CA H1047R on structure size and shape regularity was less pronounced when KMT2D was also inactivated, suggesting that the combination of these KMT2D KO in double-mutant cells attenuated cell transformation induced by strong PI 3-kinase activity. The data is convincing.

Figure 3 PIK3CA H1047R and KMT2D KO effect on collective cell migration

Looks correct, without evident difference between H1047R and the double mutant.  

The data is convincing.

Figure 4 despite anything points at an active role of KMT2D on migration, the authors examine KMT2D and PIK3CA controlled genes .

They selected in the lists of genes deregulated by either KMT2D mutation or PIK3CA H1047R mutation 11 that showed significant variations in the set of four cell lines (Figure 4).

Only two out of the 11 remaining genes showed significant variations in the set of four cell lines (Figure 4). These two genes were first, PLAUR that encodes the urokinase plasminogen activator receptor (uPAR), a glycolipid anchored protein of the cell surface involved in migration and invasion, and second, ARPC5L, that encodes one of the two paralogous ARPC5 subunits of the cytosolic Arp2/3 complex. Both CA mut and KMT2D deletion regulated PLAUR and ARPC5L expression.

The results are excellent.

Figure 5,6 Consequences of depleting ARPC5L in collective migration; no effect upon ARPC5L depletion.

In contrast, when the persistence of directional migration, was examined, PIK3CA H1047R cell line displayed increased persistence, which was even stronger when the PIK3CA mutation was combined with KMT2D KO. Importantly, the increased migration persistence in the double-mutant cell line was abolished when ARPC5L, but not ARPC5, was depleted (Figure 6D).

In this section, an appropriate video would help to clarify the conclusions on migration persistance, indeed in video S5 the authors conclusions on PIK3CA H1047R cell line displaying increased persistence, which was even stronger in double mutant cells. We dont see the effect of ARPC5L depletion on PIK3CA H1047R or double mutant cells persistent directed migration. This section needs some more improvement.

As stated in Mar 18, 2022 https://doi.org/10.7554/eLife.69229

“persistent movement requires stable polarization”; might be you can test whether ARPC5L depletion affect the persistent localization of MTOC and golgi apparatus at the front half  of the cells (indicator of polarization).

Minnor,

-on line 266 correct “tranformed”

-Last Video S5 is not compared the effect of ARPC5L depletion to that of 331 depletion of its paralog ARPC5, I thinks it represents migration persistance.

Author Response

Thanks for your comments. Please find the attached reply

Round 2

Reviewer 2 Report

Comments and Suggestions for Authors

I accept the comments by the authors and reommend publication

Author Response

There is no specific comment that should be addressed.

Reviewer 3 Report

Comments and Suggestions for Authors

No more comments. The authors addressed my concerns satisfactory.

Author Response

(The authors gave the same response as above.)

Reviewer 4 Report

Comments and Suggestions for Authors

The paper has been improved but still some aspects need to be corrected.

In line 220, table S2 does not show what is said. Same table is mentioned again in line 280. The table itself is unclear.

In line 222-3, a previous article reports the action of constitutive active PI3Kalpha in cell cycle progression, explaining the lower mean of PI3KH1047R in 1% GF.

In line 256, the fact that PI3Kalpha also increases cell size shall be mentioned as it would also explain this result.

In line 322-3, the search for synergy shall be explained as this synergy is not seen in p287-8, 252-4, 247-8.

In line 342-50, the data referring to migration persistence and MDS is only clearly different between car and double mutant (video S5 versus Figure6).
